# Precision Medicine for Rheumatoid Arthritis: The Right Drug for the Right Patient—Companion Diagnostics

**DOI:** 10.3390/diagnostics11081362

**Published:** 2021-07-29

**Authors:** Richard Thomas Meehan, Isabelle Anne Amigues, Vijaya Knight

**Affiliations:** 1Department of Medicine, Rheumatology Division, National Jewish Health, Denver, CO 80206, USA; AmiguesI@njhealth.org; 2Immunology Department, Children’s Hospital, Aurora, CO 80045, USA; Vijaya.Knight@childrenscolorado.org

**Keywords:** companion diagnostic test (CDx), biomarkers in RA, biologics for RA, precision therapeutics for RA, RA drug responsiveness biomarker

## Abstract

Despite the growing number of biologic and JAK inhibitor therapeutic agents available to treat various systemic autoimmune illnesses, the lack of a validated companion diagnostic (CDx) to accurately predict drug responsiveness for an individual results in many patients being treated for years with expensive, ineffective, or toxic drugs. This review will focus primarily on rheumatoid arthritis (RA) therapeutics where the need is greatest due to poor patient outcomes if the optimum drug is delayed. We will review current FDA-approved biologic and small molecule drugs and why RA patients switch these medications. We will discuss the sampling of various tissues for potential CDx and review early results from studies investigating drug responsiveness utilizing advanced technologies including; multiplex testing of cytokines and proteins, autoantibody profiling, genomic analysis, proteomics, miRNA analysis, and metabolomics. By using these new technologies for CDx the goal is to improve RA patient outcomes and achieve similar successes like those seen in oncology using precision medicine guided therapeutics.

## 1. Introduction

Despite major therapeutic advances in Rheumatoid Arthritis (RA) treatments due to biologics and small molecule pathway inhibitors over the last 2 decades, most patients fail to achieve remission despite evidence that earlier and more aggressive pharmacologic therapy improves outcomes [1]. The oncology world has been revolutionized by precision medicine where effective individualized targeted drug treatments are now available for specific types of cancers based upon tumor-specific markers or gene rearrangements [2]. A review article by Anderson has also proposed defining asthma subtypes (endotypes) based upon different molecular mechanisms to improve treatment responses [3]. Unfortunately, in contrast to oncology, there are no validated companion diagnostic tests (CDx) that can accurately predict which specific conventional disease-modifying anti-rheumatic drugs (cDMARDs), biologic DMARDS (bDMARDs), or small molecule drugs which target JAK pathways will be most effective for a specific patient with RA or other immune-mediated inflammatory diseases (IMID). 

Improved RA disease control with bDMARDs or small molecule drugs has resulted in major health sector cost savings from reduced need for total joint arthroplasties, remaining productive in the workforce longer, and a reduced likelihood for long term care and confinement when mobility and independence are no longer possible [4]. Despite studies showing improved outcomes in patients started on bDMARD or small molecule drugs, the approach to treat DMARD naïve RA patients remains to start a cDMARD such as methotrexate (MTX) and switch therapeutics or add an additional drug only if the patient has not responded after 3 months. Another switch may be needed after an additional 3-month trial with the new regimen often repeating this 3-month cycle until low disease activity (LDA) is achieved. 

The current trial and error approach with so many different bDMARDS and small molecule drugs after an RA patient fails, or is intolerant to methotrexate, is a huge US cost burden approaching 17 billion dollars per year. Three tumor necrosis factor-alpha inhibitor (TNFi) drugs, etanercept, adalimumab, and infliximab, which are commonly prescribed for patients with RA, psoriatic arthritis, spondyloarthritis, and inflammatory bowel disease, are among the top 10 drugs with the highest costs in the US. The lack of a drug-responsive biomarker for IMIDs allows US insurance carriers to select their preferred bDMARD or JAK inhibitor-based upon their acquisition costs or rebates rather than a physician or patient preference [5]. Despite therapeutic advances provided by TNFi therapies, 1/3 of RA patients fail to achieve LDA with this class of bDMARD [6]. Therefore, a specific patient may be forced to fail 3 or more different TNFi biologics before their insurance carrier will authorize a class switch to an agent more likely to be beneficial. 

In a large US registry (CORRONA) involving 2242 RA patients, there was no difference in response rates for patients receiving 3 different TNFi (adalimumab, infliximab, or etanercept), however, the response rates were diminished among patients who switched TNFi biologics compared to the biologically naïve patients [7]. This registry was from both academic and private practices of over 200 US rheumatologists but was not a prospective study, therefore, patients were not randomly assigned to receive a specific TNFi. The choice of bDMARD in the CORRONA registry was most likely greatly influenced by the patient’s insurance carrier rather than physician preference and it is unknown if a class switch after the first failed TNFi might have resulted in improved control by a non-TNFi bDMARD or JAK pathway inhibitor. In the United Kingdom (UK), an ongoing multicenter study and 9 industry partners, Maximizing Therapeutic Utility in RA (MATURA), will utilize bio banked synovial tissue or blood samples to identify potential biomarkers from early RA patients prior to treatment to determine if drug responsiveness to MTX or several biologics can be identified [8]. An additional prospective study involving industry partners and 140 investigators out of the UK, (RA-MAP Consortium), is an additional ongoing study to more accurately stratify patients based upon clinical features and drug responsiveness using a multi-omics approach [9].

Since millions of RA patients fail to achieve LDA from first-line cDMARDs such as methotrexate, up to 30% will need to advance to a bDMARD or small molecule drug in addition to MTX or as monotherapy due to MTX intolerance or lack of responsiveness [10]. As noted in Table 1, there are currently 5 different classes of FDA-approved biologics and JAK pathway inhibitors for the treatment of RA and various other IMID in the United States, in addition to a growing number of biosimilars for bDMARDs. Rheumatologists currently have multiple therapeutic options for treating RA patients including; 5 tumor necrosis factor inhibitors, 4 anti-cytokine biologics, anti-B cell monoclonal antibodies (rituximab), a T cell co-stimulation modulator (abatacept), or 3 small-molecule JAK inhibitors (tofacitinib, baricitinib, and upadacitinib), in addition to a growing number of biosimilars. An individual RA patient may therefore receive over a dozen different biologics or JAK inhibitors for a 3-month trial of each drug to try and achieve LDA. These frequent and expensive medication changes will place RA patients at increased risk of drug toxicity and/or irreversible joint damage from years of poorly controlled disease before the optimum drug is identified. It is also likely that the disease phenotype can change over time. The likelihood of achieving remission may lessen the longer an RA patient remains with poorly controlled disease activity and becomes more refractory to therapeutic agents. This has also been observed when an effective therapeutic agent is stopped due to co-morbid conditions, including hospitalization, infection, or surgery, with a subsequent disease flare, that patient may no longer be responsive to that same agent when re-introduced. The purpose of this review article is to identify the more promising laboratory-based platforms which may facilitate a precision medicine approach in RA management which is urgently needed to guide rheumatologists in selecting the optimum drug for an individual patient at the onset of their disease. 

## 2. How Often and Why Do RA Patients Change bDMARD and Small Molecule Drugs?

To better understand how often and why RA patients and their rheumatologists make changes in their Biologic and small molecule drug therapies, we utilized our Electronic Medical Record (EMR, Allscripts^TM^, Chicago, IL, USA, city, state abbr. (if USA or Canada), country) linked National Jewish Health Research Health Database involving over 1300 RA patients seen at our academic institution between 2008 and 2016. Detailed clinical information from 239 RA patients without RA-related lung disease was reviewed. All patients were followed >12 months, had multiple rheumatology visits, had their dictated clinic notes reviewed by 1 of 3 rheumatologists, and placed into a REDCap database in an Institutional Review Board approved research study (HS 2978). Very few of these patients had a new-onset illness, since prior to their first rheumatology visit, they had an average disease duration of 9.7 ± 9 years, 32% had erosive disease, and 23% had deformities. Each RA patient had received 2.5 ± 2 DMARD drugs prior to their first visit.

Most of these patients changed biologic or small molecule drug therapy every 2–3 years. As noted in Figure 1, drug therapy was changed primarily for lack of efficacy (44%) and/or adverse events (20%), rather than because of developing a new medical problem, disease improvement, medication cost, insurance issues, or patient preference. However, the bDMARD prescribed by the treating rheumatologist was often based upon approval of a preferred therapeutic agent by the patients’ insurance carrier’s preference.

## 3. Commonly Used Biomarkers in Rheumatology

Currently, many laboratory and imaging modalities correlate with RA clinical disease activity. The most widely used wet biomarkers include erythrocyte sedimentation rate (ESR) and C reactive protein (CRP) levels, which are nonspecific markers of active inflammation and often correlate with active synovitis in many, but not all, RA patients. Serologic tests such as rheumatoid factor (RF) and the more specific anti-citrullinated protein antibodies (ACPA) are usually associated with more aggressive disease including deformities, erosions seen on imaging and extra-articular disease manifestations. Even more sensitive than the clinical examination of active joint inflammation which makes up the DAS 28 score used to document disease activity, are magnetic resonance imaging (MRI) and ultrasound examinations to detect active synovitis [11,12,13]. Recently a multi-biomarker disease activity (MBDA) panel of peripheral blood cytokines, chemokines, and certain proteins has become commercially available (Vectra DA) for detection of active RA disease but has not been validated for predicting responsiveness for a specific drug among RA patients [14]. In a nationwide registry of RA patients in the UK, a correlation was noted between RF and ACPA positivity and reduced responsiveness to 3 TNFi drugs (Infliximab, etanercept, and adalimumab), but these biomarkers were not useful at predicting responsiveness to a specific TNFi drug in an individual patient [15]. 

Blood-based biomarkers have also been used for years by rheumatologists to identify those patients at higher risk for drug toxicity. These have included measuring glucose 6 phosphate (G6PD) levels prior to dapsone administration to prevent hemolytic anemia, and thiopurine methyltransferase (TPMT) levels to identify individuals at higher risk for developing azathioprine (AZA) toxicity. Drug levels are also measured to confirm compliance, and pharmacogenomics testing can be used to identify patients with gene expression alterations which can influence drug metabolism to help identify patients at greater risk of drug toxicity [16]. The expense of pharmacogenomics testing, which is usually not covered by most US insurers, and the lack of validation for the utility of this information in therapeutic decision-making for most patients with IMIDs, may explain why it is currently rarely used by most US rheumatologists. 

Antidrug antibody testing (ADA) has also been performed to identify why specific RA patients have lost responsiveness to a previously beneficial bDMARD. Human anti-chimeric antibody levels (HACA) have been measured in patients who previously had good disease control from the chimeric TNFi mAb, infliximab, to determine if their RA has progressed and is no longer responsive to this TNFi, or if the patient has developed HACAs. In a report of TNFi naïve RA patients, Bernucci et al. observed ADAs after 6 months of treatment in 33% of those receiving adalimumab, 12% of those on etanercept, but surprisingly in only 10% of those receiving the chimeric mAB, Infliximab [17]. They also noted an association between patients who developed IgG4 ADAs and adverse events, in addition to worsening disease control as documented by higher measured DAS 28 scores. In a meta-analysis of 17 studies involving 865 RA patients, another group reported that the development of ADAs against infliximab, and adalimumab was associated with a 68% reduced drug responsiveness rate, but that ADAs were less likely to develop among patients receiving concomitant cDMARD [18]. 

## 4. Peripheral Blood as a Source of CDx Biomarkers

Serum proteins, cytokines, and chemokines have been measured for years in RA patients to better understand immunopathogenesis, prognosis, disease susceptibility and to document systemic disease activity. Peripheral blood is easily accessed, processed, and cryopreserved and is also routinely obtained in all RA patients to monitor for DMARD toxicity. Therefore, it is an appealing source of potential biomarkers which may lead to a future validated CDx to help predict drug responsiveness in an individual patient. In addition to the MBDA previously discussed, investigators have used newer methods to screen for potentially useful drug-responsive biomarkers. Hueber et al., reported sera biomarker results among 3 ethnically diverse groups of RA patients from North America *n* = 29, Sweden *n* = 43, and Japan = 21, who were placed on etanercept [19]. They identified a 24-biomarker panel using ELISA and a cytokine multiplex platform which yielded a positive predictive value of etanercept responsiveness of 58% to 72%, and a negative predictive value of 63–78%. Blaschke et al., utilized 2D gel electrophoresis and western blot technology to identify 4 out of 55 proteins that were elevated in etanercept responders compared to non-responders. They reported that haptoglobin-alpha1, haptoglobin-alpha2, vitamin D binding protein, and apolipoprotein C-III were upregulated in etanercept responders before therapy was initiated [20]. Another group has utilized a whole blood biomarker panel and a machine learning-based algorithm (PrismRA) to aid in the identification of TNFi non-responders among bDMARD naïve RA patients using a combination of microarray gene expression, single nucleotide polymorphisms combined with several clinical features [21]. 

Lymphocyte phenotyping has also been studied to determine if this information might be a good predictor of drug responsiveness in RA. Schreiber et al. reviewed results from 25 separate studies in RA involving MTX, several TNFi, tocilizumab, abatacept, and rituximab and concluded that this information lacked sufficient predictive value to be of clinical utility regarding drug responsiveness [22]. However, there are currently newer technologies that may hold great promises such as mass cytometry, which is a combination of multi-parameter flow cytometry and mass spectrometry (CyTOF^®^). This is a promising tool for high throughput analysis of cellular biomarkers and analysis of signaling pathways at the single-cell level [23]. Mass cytometry has also been utilized to demonstrate changes in cellular composition following TNFi treatment in RA. In preliminary experiments, Nair et al. utilized the CyTOF^®^ platform to analyze immune cell-specific key signaling pathways that are activated in response to TNFα and that are modulated in response to successful TNFi therapy [24]. Utilizing peripheral blood samples, they observed differences in the basal activation level of the TNFα pathway and the relative cellular composition between TNFi treated and pre-treatment samples. Although such studies are in their infancy and further utility of CyTOF^®^ for prediction of response to therapy is yet to be demonstrated, it promises to be a powerful discovery tool for analysis of cellular pathways that may then be used to understand immunological changes in response to DMARD, bDMARD, and other therapies, and therefore lead to future CDx development. 

## 5. Autoantibody Profiling

The spectrum of autoantibodies that are associated with either the diagnosis of RA or disease progression has expanded beyond RF. The presence of ACPAs including anti-CCP are associated with more severe RA and articular destruction. While anti-CCP has long been used to evaluate disease severity and improve diagnostic accuracy, other ACPAs such as anti-citrullinated fibrinogen [25,26], anti-citrullinated collagen [27], and anti-citrullinated vimentin [28], among others have been shown to induce pro-inflammatory mediators such as TNFα and enhance neutrophil-mediated inflammation. Antibodies directed toward carbamylated antigens, observed in both ACPA positive and negative patients, have been shown to correlate with disease severity [29,30]. Other autoantibodies under investigation for their potential in diagnosis or monitoring include; anti-hinge antibodies [31,32] that are generated from the cleavage of IgG molecules by increased levels of endogenous proteases such as MMPs, and anti-acetylated protein antibodies [33]. As acetylation, an enzymatic post-translational modification of lysine occurs in both human and bacterial cells, it has been suggested that these antibodies may provide an understanding of the link between microbiome dysbiosis and the development of RA [34]. Analysis of these newer classes of autoantibodies is currently limited to understanding their role in the pathogenesis of RA, however, such studies may lead to their potential as biomarkers for disease staging and possible therapeutic decisions in the future.

## 6. Synovial Fluid as a Source of CDx Biomarkers

Synovial fluid (SF) is a promising source of valuable biomarkers to predict drug responsiveness in RA since peripheral blood, contains tens of thousands of different proteins over a very large dynamic range, released from multiple extra-articular sites or even from the gut microbiome which contains 10^14^ organisms [35]. Therefore, measuring peripheral blood proteins will not likely reflect changes in the synovial fluid or intra-articular structures. Prior clinical studies have also demonstrated a poor correlation between key regulatory cytokines in the peripheral blood compared to SF levels in RA, OA, and traumatic arthritis [36,37,38,39]. In a study by Wright et al. from 42 RA patients, they found that 12 SF cytokines were much lower in the peripheral blood than when measured simultaneously in the SF and correlated less well at predicting TNFi responsiveness [40]. The peripheral joints are also the primary site of RA disease expression and active inflammation. Furthermore, since cartilage has limited ability for repair and regeneration, irreversible joint damage can result in permanent disability from the degradation of chondrocytes and extracellular matrix due to pro-inflammatory cytokines and degradative proteases in addition to pannus invasion [1]. The hallmark of inflammatory SF is reduced viscosity, increased volumes, and an increase in pro-inflammatory cytokines which are catabolic to chondrocytes and the extracellular cartilage matrix. 

Proteomic analysis has also been performed on SF and serum proteins using Mass Spectrometry (MS) to investigate possible mechanisms of RA disease activity and changes with TNFi therapy but has not yet been demonstrated to accurately predict drug responsiveness [41,42]. 

Another advancement in immunology that may facilitate biomarker discovery in SF are the multiplex technologies using fluorescent beads and flow cytometry methodology, which allows investigators to measure many cytokines, chemokines, and proteins simultaneously from volumes < 200 mcL [35,36,37,38,39,40]. Access to SF samples, however, has been challenging for many researchers as in the absence of image guidance, it may be technically difficult to obtain SF in many patients due to small volumes, high viscosity, and obstruction of the needle with synovial tissue or obscured landmarks due to obesity. Published studies indicate that in the absence of image guidance physicians may actually miss the intra-articular knee space for injections by up to 25% of attempts [43]. However, ultrasound localization of SF and other technologies may allow easier sampling of SF in the future [44,45,46].

## 7. Synovial Tissue as a Source of CDx Biomarkers 

Synovial tissue sampling either via arthroscopy or with ultrasound-guided needle biopsy has demonstrated different RA disease phenotypes which may explain why some patients are more responsive to certain biologic classes of therapeutic agents than others, based upon the infiltration of specific cell types. Ninety-seven synovial biopsies obtained from pre-treated RA patients revealed that lymphoid aggregates were noted in 67% of infliximab responders, whereas those findings were only observed among 38% of non-responders [47]. In another study that measured B cells after rituximab therapy, the authors reported depletion of B cells in the peripheral blood but not in the synovium [48]. Preliminary studies using mass cytometry comparing immune cells isolated from RA or healthy control synovial tissue have been informative in defining changes in specific inflammatory cell populations and mapping these cell populations such as pro-inflammatory monocytes or activated synovial fibroblasts to specific inflammatory mediators [49].

Gene expression within the synovium has been reported by Hogan et al., who studied gene expression via quantitative polymerase chain reaction (qPCR) and demonstrated that 20 RA patients who were refractory to TNFi therapy had an altered synovial tissue gene expression pattern among rituximab responders vs. non-responders [50]. Using supernatants from synovial fibroblasts from RA patients, healthy donors, or bDMARD treated RA patients on an in vitro human chondrocyte culture model, Adreas et al., identified overexpression of 110 related genes which were catabolic to chondrocytes, whereas anabolic mediators were under-expressed [51]. Similar technologies might not only identify unique pathways for RA disease expression but also reveal potential novel targets for therapeutic agents, or biomarkers to identify drug responsiveness in individual patients. 

While synovial biopsy provides an exciting new opportunity to explore mechanisms of disease expression with newer molecular tools to phenotype patients into various immunologic pathways, there is limited evidence this can prospectively categorize patients into specific drug responders vs. non-responders currently. The time needed to master this technology as well as the cost of high-fidelity Ultrasound equipment and the current low reimbursement for these procedures in the US makes this an impractical test for an illness affecting 1% of the population. 

## 8. Potential Additional Promising New Technologies for CDx

Proteomics: Methodologies such as mass spectrometry (MS) and liquid chromatography (LC) enable not just quantification of large numbers of proteins and peptides, but also analysis of their structure, function, modifications, and interactions in a variety of sample types. MS, LC-MS, and Matrix-assisted laser desorption/ionization time-of-flight (MALDI-TOF) are some of the methods that have been utilized for proteomic studies in RA. Samples utilized for analysis have included serum, plasma, synovial fibroblasts, fluid or tissue [52,53], peripheral blood mononuclear cells [54], monocyte/macrophages, and neutrophils. The focus of proteomic studies has been largely to identify proteins or peptides that are unique to RA patients compared with healthy individuals. Several groups have published data from proteomic studies performed on fibroblast-like synovial cells and have identified proteins that are unique or are differentially expressed in RA samples. These proteins include structural and enzymatic proteins, some of which have been identified to be autoantigens. Although proteomic studies have largely focused on diagnostic markers for RA, monitoring disease activity and severity, they have also been utilized to assess treatment response. In a study of 20 RA patients treated with triple therapy including MTX, leflunomide, and TNFi following the failure of MTX and leflunomide combination therapy, Chen et al. demonstrated 51 differentially expressed peptides in the serum of responders that correlated with response to TNFi, 3 of which were predictive of a good response [55]. In a smaller study of 10 RA patients, Dwivedi et al. compared peptide profiling of pre-treatment and Infliximab treated serum samples and showed that a robust response to infliximab was associated with downregulation of several TNFα -regulated proteins [56]. These preliminary studies indicate that proteomic profiling of RA samples has the potential to identify proteins or peptide panels that, in the future, may be predictive of responsiveness to different bDMARDS. 

## 9. Additional Genetic Based Platforms 

Kim et al. reviewed the results of genome-scale transcriptome profiles from RA synovial tissue in previously published studies. Given the large numbers of deferentially expressed genes (DEG), they utilized machine learning to determine among the 62 RA patients in these studies, who had drug responsiveness data to Infliximab, identifying some DEGs involved in various signaling pathways which may have accounted for this association [57].

Measuring specific miRNA (microRNA) levels is another potential biomarker for drug responsiveness. As reviewed by Latini et al., they report prior studies which have demonstrated alterations in specific miRNA levels following treatment with various bDMARDs [58]. While several miRNAs were upregulated in the serum after TNFi therapy, there is no pretreatment miRNA signature that, at this time, accurately predicts drug responsiveness in RA or IMID patients. However, since miRNAs are stable in body fluids, can be accurately quantified, and are known to regulate gene expression involved in drug metabolism and in numerous immunologic pathways, this emerging field of pharmacoepigenomics is an exciting field that may be of value if a validated CDx can predict which specific bDMARD or JAK inhibitor drug responds best in an individual patient. 

Whole-genome microarrays were utilized by Derambure et al. in a prospective trial of abatacept + MTX in RA to identify differential gene expression among the 36 responders compared to the 19 non-responders [59]. They identified 87 transcripts that were differentially expressed among the responders compared to non-responders. Canet et al., investigated which among 47 single-nucleotide polymorphisms might be associated with TNFi responsiveness using a registry of 1985 TNFi treated patients [60]. They identified a significant correlation with improvement in DAS28 after TNFi with the CYP3A4 and CYP2CP variants. Folkersen et al. used transcriptomics with high throughput RNA sequencing, genomics with genotyping microarrays, and proteomics from banked blood samples of 59 RA patients before being placed on TNFi [61]. They reported a sensitivity of 0.73 and specificity of 0.78 for improved DAS28-CRP scores after 3 months on TNFi for 2 proteins, 2 SNPs, and 8 mRNA biomarkers. 

Metabolomics is another emerging technology that could have potential value as a CDx. As reviewed by Gupta et al., several thousands of these small molecules in various tissues and fluids can be measured using LC-MS, gas chromatography-MS (GC-MS), or nuclear magnetic resonance spectroscopy (NMR) [62]. This information may provide insight into various immunometabolic profiles, or pathways in different RA phenotypes, which might have predictive value in drug responsiveness. Studies are ongoing to investigate if the metabolomics profile can predict TNFi responsiveness [63]. Given the potentially large data generated from these various platforms, complex artificial intelligence (AI) algorithms or other powerful analytic tools may become necessary to confirm the sensitivity and specificity of metabolomics profiles to validate a clinically useful CDx for individual patients with RA or IMID. 

## 10. Conclusions

The FDA usually will approve a new biologic or small molecule drug for RA if clinical studies demonstrate an improvement in ACR20 scores compared to placebo or MTX, or non-inferiority in the case of biosimilars to the original biologic. These trials involve large numbers of patients who often have not failed numerous cDMARDs or even bDMARDs. To be of clinical value, a CDx test for drug specificity for an individual patient with an autoimmune illness must be validated to provide good predictive value for responsiveness to a specific drug in an patients with similar phenotypes. This will be difficult given the heterogeneity of disease phenotypes as well as drug responsiveness, even within a specific IMID such as RA, and will therefore require a large cohort of patients [2]. Therefore, such an adequately powered study will also be very expensive and require uniform sample handling and processing, depending upon the biomarker or omics approach used. Analysis from this large trial involving numerous potential biomarkers will involve multiple centers and possibly the utilization of complex computation analysis such as AI and bioinformatics. To reduce variability from different labs performing analysis, the initial validation trial should ideally be performed by a single lab. To conduct such an expensive trial in the US, the most likely funding source would be NIH, DOD, or a very well-endowed charitable foundation as the Gates foundation. Pharmaceutical companies may not have the financial incentive to validate a CDx that could suggest fewer patients would actually be prescribed their drug rather than the current “trial and error” approach. 

Another potential limitation of an accurate CDx is the confounding effects that concurrent therapies may have on test results, especially cytokine levels. This is a compelling reason why it would be ideal to obtain a CDx before the patient has started on their first immunomodulatory drug. Even if a patient has discontinued a drug with minimal clinical efficacy before the test is performed, it is still possible that the drug might have altered the immunologic pathway of the disease expression and therefore the test results may reflect those alterations rather than the patient’s ability to respond to a specific drug. It may also improve the predictive value of a test to include unique clinical or phenotypic features of the patient in the analysis as was included in the platform by Mellors et al. [21]. 

The ideal CDx would need to be relatively inexpensive, available as a point of care test in the physician’s office as a blood-based assay, and require minimal sample preparation or manipulation. This wet biomarker assay could be based upon some of the above technologies including a cytokine/chemokine, a genetic or omics-based assay of a validated panel of biomarkers. It is also possible that to identify the key targets of interest for inflammatory arthritis, investigators may need to first employ more technically difficult procedures such as Ultrasound-guided synovial biopsies for cytokine or gene expression, or for obtaining SF. However, for a CDx to be of great value, these initial pilot studies will need to be validated against a surrogate peripheral blood-based biomarker panel that is affordable and requires simplified sample handling methods. A validated CDx would revolutionize the care of patients with autoimmune illnesses by allowing the judicious use of these very expensive biologics and small molecule drugs to enhance a personalized medicine approach and allowing rheumatologists to prescribe the best drug for an individual patient early in the disease process.

## Figures and Tables

**Figure 1 diagnostics-11-01362-f001:**
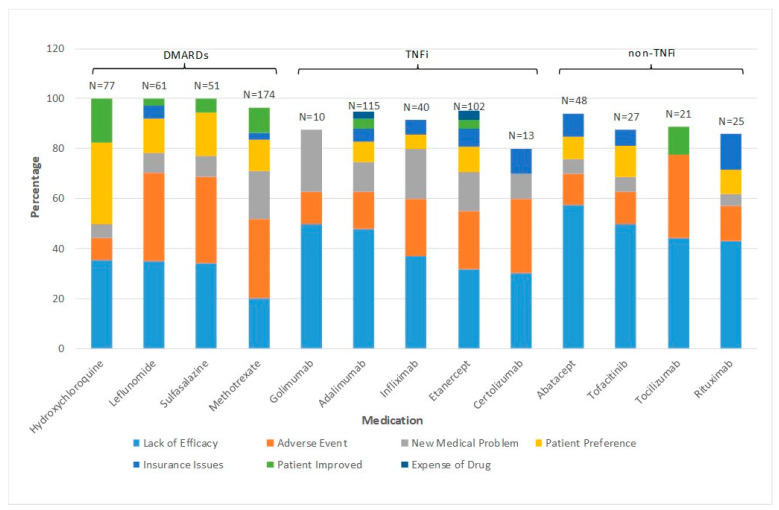
Results of reasons why RA patients change biologic or small molecule therapy within one academic practice in the US between 2008 and 2016.

**Table 1 diagnostics-11-01362-t001:** FDA approved biologic and small molecule drugs for the treatment of RA diseases and other IMID in the US.

TNFi	Mechanism	Auto-Immune Disease Indications
Etanercept	fusion protein-TNFr	RA, PsA, AS, JIA
Infliximab	Chimeric anti-TNF	RA, PsA, AS, Crohn’s, UC
Adalimumab	Anti-TNF mAB	RA, PsA, AS, JIA, UC
Certolizumab Pegol	pegylated anti-TNF	RA, PsA, AS, Crohn’s, UC
Golimumab	anti-TNF mAB	RA, PsA, AS
**Anti-cytokines**		
Anakinra	IL 1 ra	RA, JIA
Canakinumab	anti-IL 1 mAB	Systemic JIA
Tocilizumab	anti-IL6 mAB	RA, GCA, JIA
Sarilumab	anti-IL 6 receptor	RA
**Anti B Cell**		
Rituximab	chimeric anti-CD 20	RA, GPA, MPA
**T Cell Costimulation modulator**		
Abatacept	CTLA-4 fusion protein	RA, PsA, JIA
**Small Molecule**		
Tofacitinib	JAK inhibitor	RA, PsA, UC, JIA
Baricitinib	JAK inhibitor	RA
Upadacitinib	JAK inhibitor	RA

AS—Ankylosing Spondylitis, GCA—Giant Cell Arteritis, GPA—Granulomatosis with polyangitis, IL1ra—Interleukin 1 Receptor Agonist, JAK—Janus Kinase, JIA—Juvenile Idiopathic Arthritis, mAB—monoclonal antibody, MPA—Microscopic Polyangitis, PsA—Psoriatic Arthritis, RA—Rheumatoid Arthritis, TNFi—Tumor Necrosis Factor Inhibitors, TNFr—TNF Receptor and UC-Ulcerative Colitis.

## Data Availability

The Data used to generate the results in Figure 1 are in a REDCap database at National Jewish Health and contains PHI so it is considered confidential except for the investigators who contributed or analyzed the data.

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
