# Peer review of "Precision Medicine for Rheumatoid Arthritis: The Right Drug for the Right Patient—Companion Diagnostics"

_diagnostics, 2021, doi:10.3390/diagnostics11081362_

Round 1

Reviewer 1 Report

Meehan RT et al. discuss the literature regarding the use of “companion diagnostic” tests for the stratification of RA patients to treatments best able to reduce disease activity.  The authors paint with a broad brush but raise several key points including (at Ln 364) the inclination for pharmaceutical companies to resist a diagnostic that might reduce the potential use of their product.  The onus is thus on clinicians and non-pharma scientists to advance this area of research.  It would improve the manscript if the concluding paragraph of the Introduction succinctly explained the rationale for the paper.  Several large studies are currently in progress to facilitate patient stratification (eg PMID: 28165532, PMID: 29213124) to which the authors should refer.

1>

At several points a comparison is made to oncology precision medicine however a better comparison might be that of allergy research from which arose the term endophenotype, shortened to endotype, for treating patients based upon underlying mechanisms of disease [PMID: 18805339].

2>

At line 23 the authors state that ‘most patients fail to achieve remission’.   The remission achieved may not be permanent but the majority of patients have an initial response to biologics in combination with a DMARD.  Did the authors mean most monotherapies fail?

3>

At Line 71 the authors refer to ‘5 TNF inhibitors, 4 anti-cytokine inhibitors … anti-T cell therapy…’

TNF is a cytokine but this phrasing seems to indicate that it is not.  Abatacept is not an anti-T cell therapy in the same way that Rituximab is.  Abatacept prevents the 2nd costimulatory signal from an APC that activates the T cell.

4>

Does the heading at Ln 95 require a question mark?

5>

The authors have included primary data in their paper which is uncommon for a review.  It is unclear whether the treatments are combination or monotherapy.  Changing the Y-axis to terminate at 100% would improve the appearance of the graph.  EMR and IRB are not defined.

6>

Ln 199 “Autoantibody profiling: …” This form of formatting is not found elsewhere in the manuscript.

7>

At line 211 ”As acetylation, an enzymatic post-translational modification of lysine occurs in both human and bacterial cells, it has been suggested…” could be better expressed as ”As acetylation, an enzymatic post-translational modification of lysine that occurs in both human and bacterial cells, it has been suggested…” OR ”As acetylation, an enzymatic post-translational modification of lysine, occurs in both human and bacterial cells it has been suggested…”.

8>

Line 221 The phrase “over a 12-fold dynamic range” is fairly specific but no reference is cited.  Did the authors mean 12 orders of magnitude – from femtogram to milligram?

9>

Infliximab is abbreviated at Ln 46 (IFX) however it continues to be used in full form throughout.

10>

The authors abbreviate ultrasound as US.  As ultrasound is a single word the shortening is not overly helpful and can be confused with US (United States) used elsewhere in the manuscript.

11>

Reference 19 requires amendment (“19.   19”), as does reference 21 (“? vol and pages”), 51 (unclosed parentheses) and 52 (underlined journal name).  The number of authors listed for each references does not follow a pattern; the authors should use the Journal style.

12>

Please proofread the manuscript carefully and ensure specific points are properly referenced.

Author Response

I have attached my responses to each of reviewer number 1 very helpful comments and suggestions which have improved this content of this manuscript (style and composition suggestions) as well as the suggested 3 excellent new references which we have added. Very much appreciated. R Meehan

Reviewer 2 Report

First of all, the title of the journal is DIAGNOSTICS! Diagnostics mean something emerged from clinical phenotype, imaging studies supported if possible with laboratory findings (i.e  a triad in which the apex is clinical phenotype). This triad, at the end of the day leads to DIAGNOSIS. Anyway, authors did a review study and they utilized their findings from other agencies dealing with RA. Authors discussed sampling of various tissues for potential lack of a validated companion diagnostic and review early results The Federal Food and Drug Administration, is generally considered as the most powerful decision maker (i.e. who may thereby be led into factual or other morbid errors).

Authors wondered and asked

How often and why rheumatologists make changes in their drug prescriptions. RA patient may therefore receive over a dozen of different biologics or JAK inhibitors for a 3-month trial of each drug to try and achieve LDA. These frequent and expensive medication changes will place RA patients at increased risk of drug toxicity and/or irreversible joint damage from years of poorly controlled disease before the optimum drug is identified. The answers are

  1. Lack of comprehensive clinical documentation, this should be based on individualistic phenotypic characterizations and surely not on the bases of Herds of cattle.
  2. Authors need to mention the necessity of connecting the readings and results of cytokines (frequent switching of anti-rheumatic medications in long term RA patients can lead to proliferation of Cytokines and antibodies (anti-ribosomal-P and anti-N-methyl-D-aspartate receptor). These are strongly correlated to pathogenesis of neurodevelopmental disorders, especially depression in patients with chronic illnesses accompanied with pain such as RA.
  3. Also autoantibody profiling, genomic analysis, proteomics, miRNA analysis, and metabolomics, all such results have to be connected not only with the disease phenotype but  with the clinical phenotype of each patient.

Author Response

We appreciate this reviewer's comments and have included those excellent suggestions  those a potential limitations on CDx testing in our conclusion section.

Round 2

Reviewer 1 Report

The authors have adequately addressed the concerns of the peer reviewer.

As part of the final proofing process please ensure consistent formatting of heading fonts.